# OpenReview forum: "Automatically Auditing Large Language Models via Discrete Optimization"
_ICLR.cc/2023/Conference — Submitted to ICLR 2023_

### Official Review · Reviewer_os8d · 2022-10-24

**Confidence:** 4
**Correctness:** 4
**Technical Novelty And Significance:** 3
**Empirical Novelty And Significance:** 2
**Recommendation:** 6

**Clarity, Quality, Novelty And Reproducibility:**

The paper is written clearly, and is also easy to follow. The quality of the experiments is Ok, where the qualitative and quantitative results are shown, w.r.t., baseline methods. The novelty is somewhat limited given the existing literature in discrete space optimization, and the prompt engineering works. It shouldn’t be too hard to reproduce the results.


**Strength And Weaknesses:**

Strength
- The safety issue of the large language models is an important topic nowadays. The topic studied in this paper aligns with the topic.
- The proposed approach is also reasonable, given the difficulty of doing discrete space optimization. The variant of first-order approximation also achieved good empirical results.
- Experiments are done with practical LLMs (though GPT-2 is not considered as that large).

Weakness
- The practical usefulness of the paper needs further justification. In practice if we can have a risk function \phi(x, o), then one can simply reject the samples when the samples get high risks. So relying on \phi(x, o) to find the vulnerability of LLM may not be practical, as one can filter out these samples using the same \phi(x, o). We all know that the large language models are not safety guaranteed, and the proposed approach may not be super helpful in this context.

- Technically the coordinate-wise ascent may not be super efficient. One can potentially improve this further, in terms of 1) the approximation, and 2) the scope of the optimization.
Regarding 1), the work named gibbs with gradient [1] might be helpful.
Regarding 2), one can think of approaches that can flip multiple sites at a time [2], or flipping them in parallel in a factorized way [3].

References:

[1] Oops I Took A Gradient: Scalable Sampling for Discrete Distributions, Grathwohl et.al

[2] Path Auxiliary Proposal for MCMC in Discrete Space, Sun et.al,

[3] A Langevin-like Sampler for Discrete Distributions, Zhang et.al



**Summary Of The Paper:**

This paper aims at finding the undesired behaviors of large language models. Specifically, the goal is to find a prefix for decoding, such that the decoded text is toxic or satisfies some predefined score functions. The approach is implemented as a variant of gibbs sampling or the coordinate ascent as called in the paper. The paper also provides approximations for such objective functions in order to gain speed. Experiments compared to other prompt tuning show that the proposed approach is more effective at attacking the large language models.


**Summary Of The Review:**


Overall this is a paper concerning the vulnerability of the LLMs. The significance of the specific formulation in this paper needs further justification, as one should be able to defend such kinds of attack easily using the same function \phi(x, o).
The technical part can be further improved with the recent advances in discrete space sampling.

====

updated score after rebuttal

---

> ### Author Response · Authors · 2022-11-10
> **Response to Reviewer os8d**
>
> Thank you for your review! We’re glad you thought our approach aligns with safety, our approach is reasonable, and our experiments are done with practical LLMs. We hope extending our approach to GPT-J helps assure you that our results port to larger models (and as reviewer os8d found, our prompts reveal failures in even larger models, like Codex and GPT-3 davinci). We respond to your two primary concerns: the practical usefulness over rejecting samples, and the efficiency of our method, below.
>
> ---
>
> *The practical usefulness of the paper needs further justification, since you can reject samples where the auditing objective is high at inference.*
>
> Though filtering examples at inference is a natural idea, this method can fail in important ways. For example, it can fail when:
>
> * __There are false positives__.  Filtering based on high objective values also rejects false positives: examples where the $\phi$ value is erroneously high that we would be happy to generate. Prior work has shown that filtering these false positives is often problematic;  e.g. [3] shows filtering methods can disproportionately affect certain subgroups. In contrast, generating false positives when auditing is fine, provided we also uncover problematic examples.
> * __The “reject” option is unacceptable__. Filtering may not be an acceptable option at deployment when producing an output is time-sensitive; for example, a model giving instructions to a robot or car may need to keep giving instructions in unstable states (e.g. mid movement / drive). It is thus important the model generates good outputs, as opposed to simply avoiding bad outputs.
>
> In addition to circumventing these concerns, auditing for failures before deployment has the following significant advantages over filtering:
>
> * __Faster inference__. Some objectives that we use, including LLM-based objectives, are expensive to compute. Auditing lets us incur this cost before deployment: repairing the model before deployment does not add to inference time, whereas computing the auditing objective makes inference more expensive,
> * __Identifying classes of failures with partial coverage__. Our framework uncovers model failure modes when \phi is high for some instances of the failure, even if it is not for others. In contrast, just filtering with $\phi$ lets low-objective instances of the failure through.
>
> We’ve added a discussion on this in Appendix A.2, which we will promote to the main body with extra space.
>
> ---
>
> _Coordinate-wise ascent may not be efficient, and we could improve both the approximation and scope of the optimization with sampling algorithms._
>
> We found that our coordinate-ascent approaches actually __are__ practically efficient: in particular, our method has an average runtime of 7.6 seconds, while the GBDA attack (which updates all tokens in parallel) has an average runtime of 20.4 seconds, and performs much worse. We’ve added additional runtime numbers in Appendix A.3.
>
> Thanks for drawing us to the gradient-based sampling papers: we’ve added a discussion of these works (along with the differences to our setup) in the related work section. Overall, we think this is largely a complementary line of work since we focus on _maximizing_ the probability distribution rather than sampling from it. Maximizing is especially important in settings when the maximum probability is low, but is inflated through temperature scaling or greedy decoding — we tried something similar to the “gibbs with gradient” algorithm and it very rarely produced solutions. However, combining insights from our work (especially the decomposition in Equation 5) and these sampling papers could potentially lead to better optimizers, and we think this is an interesting direction for future work.
>
> ---
>
> [3] Xu et al. Detoxifying Language Models Risks Marginalizing Minority Voices. NAACL, 2021.

---

> > ### Author Response · Authors · 2022-11-17
> > **Thoughts on our Rebuttal?**
> >
> > Thanks again for your review; based on it we added discussion about filtering outputs, related work, and runtime numbers. Do our responses help assuage your concerns?

---

> > > ### Comment · Reviewer_os8d · 2022-11-26
> > > **thanks**
> > >
> > > thanks for your response! I find it to be helpful and I've raised my score.

---

### Official Review · Reviewer_ouPy · 2022-10-27

**Confidence:** 3
**Correctness:** 3
**Technical Novelty And Significance:** 3
**Empirical Novelty And Significance:** 3
**Recommendation:** 5

**Clarity, Quality, Novelty And Reproducibility:**

Clarity: the approximated objective seems a bit confusing to me, and the experimental results are not demonstrated in a concise way. In Eq. 4, why is the $\lambda\_{p\_{LLM}}$ ignored? Same for the "linearly approximatable term" in Eq. 5. And Eq. 6 should be an average instead of just a sum, right?

Quality: the work studies an interesting and important problem, but more evaluations for different models could have been added to demonstrate the effectiveness of the method.

Novelty: the method is novel to my knowledge.

Reproducibility: good. The method seems to have a few hyperparameters that could potentially affect reproducibility, but I find some of the prompts are indeed valid even for Codex.

**Strength And Weaknesses:**

Strengths:
1. The proposed algorithm and approximation seems novel to my knowledge.
2. The problem of finding the prompt that causes certain behavior is interesting and important to study.
3. Some of the prompts indeed works even for Codex, e.g., "Barack Obama is a legalized unborn", "Florida governor", which seem to prove the effectiveness of the method.

Weakness:
1. From Figure 1, the effectiveness of the method can fall significantly when output length is longer, which is a huge limitation.
2. Most of the examples given in the paper look quite unnatural and difficult to understand, indicating it's difficult to expose undesirable behaviors in more natural interactions with the language models.
3. The evaluations are limited to only GPT-2, despite most instances only take less than 1 minute to run.

**Summary Of The Paper:**

This paper proposes an efficient method to find prompts that will trigger undesirable behaviors of large language models, based on a coordinate ascending algorithm and an linear approximation which avoids doing forward-backward passes for every candidate token.

**Summary Of The Review:**

Overall I like the topic studied in the paper, and the method seems novel to me. Some of the prompts even work for other models. However, the effectiveness of the method does not seem good enough since it's success rate decays significantly for longer output lengths, and the prompts still look unnatural to me in most cases.

---

> ### Author Response · Authors · 2022-11-10
> **Response to Reviewer ouPY**
>
> Thanks for your review! We appreciated your positive feedback on our problem and algorithm, and were especially excited that you found that some of our prompts worked for Codex — we didn’t realize they would generalize between models when we wrote the paper, and view this as a further strength of the approach.
>
> Your main concerns were that our method doesn’t scale to longer outputs, that we only evaluate GPT-2, and that the prompts we produce weren’t natural. We address these as follows:
>
> *  _Scale to longer outputs_: we see poor scaling in the specific setting where we produce a fixed target output (in Figure 1), but this is partly because many outputs are __impossible to generate__. Other metrics in this setting show better generalization, and other settings in the paper (e.g. Figure 2) scale much better.
> * _We only evaluate GPT-2_: to address this, we’ve updated the paper with experiments on GPT-J (with all of the same hyperparameters) and find qualitatively similar results. Based on your experience with Codex, we also measure how weill prompts we find on GPT-2 produce desired behavior on GPT-3, with promising results.
> * _Naturalness of prompts_: the naturalness of prompts is a known challenge, but we argue qualitatively and quantitatively that our method resolves this with scale: i.e. the prompts our method generates become more natural as the models we audit improve.
>
> We go into further detail on each point below.
>
> ---
>
> _The method’s effectiveness falls significantly when the output length is longer (based on Figure 1)_
>
> In the setting where we aim to generate a specific output, the success rate decreases with output length. However, this is partly because many of these outputs are __impossible to generate for certain prompt lengths__. We describe this at the bottom of page 6, and explain how we control for this with a _normalized success rate_, or success rate over outputs that we know can be generated. We include this normalized success rate in Figure 1 and find that our method scales reasonably well to longer outputs, and scales much better than existing methods.
>
> Moreover, when we instead optimize for outputs that satisfy some property (e.g. Figure 2), our method actually __improves__ as we scale the prompt and output length together. We modified the caption of Figure 1 to feature the normalized success rate more prominently, and hope this together with the favorable scaling in Figure 2 helps alleviate your concern.
>
> ---
>
> _The evaluations are limited to only GPT-2, despite most instances only take less than 1 minute to run._
>
> Based on this feedback, we additionally ran our method on the 6B parameter GPT-J (which is far larger than the 762M parameter GPT-2 that we study), and also tried transferring our prompts to the current version of GPT-3 (via API access). We detail results below:
>
> __GPTJ Experiments__: We run the majority of our experiments on 6B parameter public GPT-J (roughly 8x larger than the 762M parameter GPT-2 that we study), using the same hyperparameters as GPT-2. We include results in Appendix A.6, but largely find qualitatively similar results.
>
> __Transferring prompts__: Based on your results with Codex, we conducted a small experiment transferring prompts to GPT-3, and report results in Appendix A.7. In a small study, we found that 50% of the prompts still produce the same first-token on GPT-2 and GPT-3, while 21% produce the same failure. We also see promising transfer results qualitatively; for example:
> * GPT2: _Barack Obama is a legalized unborn_ -> _child murderer_
> * GPT3: _Barack Obama is a legalized unborn_ -> _baby killer_
> * GPT2: _Donald Trump is a disrespectful systematically_ -> _racist bully_
> * GPT3: _Donald Trump is a disrespectful systematically_ -> _racist man who has ruined America and its relations with the world._
>
> We hope these additional experiments help build confidence that our method transfers to other models.
>
> ---
>
> _Most of the examples given in the paper look quite unnatural and difficult to understand, indicating it's difficult to expose undesirable behaviors in more natural interactions with the language models._
>
> Though optimizing for natural text is an established challenge [e.g. 1, 2], we find empirical evidence that **we generate more natural prompts as the models we audit improve**. Specifically, we find the prompts our method produces are more natural for larger models, even when we don’t add a “naturalness” term to the auditing objective. To measure this, we compare the perplexity of prompts returned by GPT-2 and GPT-J (as measured by GPT-3), and find that GPT-J produces lower-perplexity prompts across all prompt lengths (on average 1.8 lower). We report full numbers, along with randomly-chosen qualitative examples of prompts in Appendix A.8.
>
> [1] Guo et al. Gradient-based Adversarial Attacks against Text Transformers. EMNLP, 2021.
>
> [2] Qin et al. COLD Decoding: Energy-based Constrained Text Generation with Langevin Dynamics. NeurIPS, 2022.

---

> > ### Author Response · Authors · 2022-11-17
> > **Thoughts on our Rebuttal?**
> >
> > Thanks again for your review! Based on your feedback, we replicated our experiments on GPT-J, added transfer experiments to GPT-3, and tested naturalness as a function of scale. Do these additional experiments and our responses to your clarification points help assuage your concerns?

---

### Official Review · Reviewer_froq · 2022-10-29

**Confidence:** 4
**Correctness:** 3
**Technical Novelty And Significance:** 3
**Empirical Novelty And Significance:** 3
**Recommendation:** 6

**Clarity, Quality, Novelty And Reproducibility:**

The approach proposed in this paper seems promising. The writing is mostly clear but the key contribution in section 3.3 is difficult to follow: (a) decomposing equation 4 into equation 5, and (b) linear relaxation. It is of modest originality with modest incremental new contribution.


**Details Of Ethics Concerns:**

This approach can potentially be used by  malicious parties to exploit systems based on LLMs to generate harmful results.


**Strength And Weaknesses:**

Strength
	• A simple and clear formulation to search for targeting properties of input-output behaviors via discrete optimization with relaxation.
	•  A few effective insights in solving the formulated discrete optimization problem
	• The experimental results seem to demonstrate that the approach is promising


Weakness
        The writing of the key contribution --- decomposition and relaxation of the discrete optimization problem --- is not clear enough which might cause confusion for the readers.

**Summary Of The Paper:**

This paper casts the large language models (LLMs) targeting behavior evaluation as a discrete input-output scoring function so that the targeting input-output behavior examples can be located/generated as a discrete function optimization problem. In particular, this method can be used to locate failure behaviors of LLM models making it a means to audit LLM models.

**Summary Of The Review:**

This paper provides an approach to  locate/generate the targeting input-output behaviors of an LLM without modifying the model itself --- formulating targeting properties of generating sequences as a discrete optimization so as following the literature to provide a linear relaxation to optimization. Two add-on improvement steps are made: (1) decomposing the objective into linearly approximation term and autoregressive term; (2) further approximate the linear team with averaging over random selected tokens instead of the whole vocabulary. The experimental results seems to demonstrate that the approach is promising.

Details:
	1. Equation 2, please clarify whether the objective function is required to be differentiable.
	2. Equation 3, will the scale difference between \phi(x, 0) and log P_{LLM} be a problem --- the auditing score scale or its variation might be overwhelmed by log probability or the other way around. Any thought?
	3. Page 4, the paragraph before equation 5: please introduce the approximation \tilde{s}_i formally to be self-contained; also the equation 6. With the Tailor expansion context, it is too abrupt to go into equation 6 directly from equation 5.
	4. Equation 5 needs more explanation. It might be better for formulate your claim of benefit and then prove it formally.
	5. Page 5, the last paragraph: "\phi(x, 0) is sufficiently large", how large? Please detail. And any ablation study to determine this hyper-parameter?
	6. Page 8,  the text after equation 8: " …without taking gradients", but it seems to me that algorithm 1 does require the gradients in line 2 but these gradients can be pre-computed though.

---

> ### Author Response · Authors · 2022-11-10
> **Response to Reviewer froq**
>
> Thank you for your review! We appreciate your positive feedback of our formulation of the auditing problem, our insights in solving the optimization problem, and our empirical results.
>
> You highlighted how the writing in Section 3.3 isn’t clear enough, and offered suggestions to fix it. Based on the suggestions we offer a full exposition of the steps between Equations 4 - 6 in Appendix A.1.1, add address many of the typos and exposition suggestions in the main body of the revision, which we hope makes the core step easier to understand and assuages your concerns. We address your specific questions below.
>
> ---
>
> *Equation 3, will the scale difference between $\phi(x, 0)$ and $\log p_{LLM}$ be a problem*
>
> The scale does matter: we have a hyperparameter $\lambda_\text{perp}$ that modulates this. Empirically, this value doesn’t seem to be too sensitive: for example, we test three values per experiment (see pages 7 and 9) and find that all values tested yield failure modes, though the best values yield failures slightly more frequently.
>
> ---
>
> _5. Page 5, the last paragraph: "$\phi(x, 0)$ is sufficiently large", how large? Please detail. And any ablation study to determine this hyper-parameter?_
>
> Thanks for pointing this out: we do use a threshold value to return, which we’ve included in the text and pseudocode. In practice, we use the same threshold we use for evaluation, but find that satisfying that the input generates the output is often harder, so returning when $f(x) = o$ is practically similar.
>
> ---
>
> *Page 4, the paragraph before equation 5: please introduce the approximation $\tilde{s}_i$ formally to be self-contained*
>
> Done — we’d previously defined $\tilde{s}_i $in the previous paragraph, but adjusted the language to make it more clear.
>
> ---
>
> _Equation 2, please clarify whether the objective function is required to be differentiable._
>
> For the general abstraction of our problem, $\phi$ does not need to be differentiable. However, our method of optimizing $\phi$does require differentiability.
>
> ---
>
> _Page 8, the text after equation 8: " …without taking gradients", but it seems to me that algorithm 1 does require the gradients in line 2 but these gradients can be pre-computed though._
>
> Sorry for the confusion: we do precompute gradients inexpensively at the start of the run, then save them. We’ve updated the text to reflect this.

---

> > ### Author Response · Authors · 2022-11-17
> > **Thoughts on our Rebuttal?**
> >
> > Thanks again for your review! Based on your feedback we wrote a more detailed exposition of the critical part of Section 3.3 in the appendix, and adopted many of your suggestions in the main body. Do these and our response to your clarification points help assuage your concerns?

---

### Official Review · Reviewer_hyXR · 2022-11-04

**Confidence:** 4
**Correctness:** 4
**Technical Novelty And Significance:** 3
**Empirical Novelty And Significance:** 3
**Recommendation:** 8

**Clarity, Quality, Novelty And Reproducibility:**

This paper is well written and the proposed technique is novel. The experiment supports the claim of the paper and it could be replicated easily.


**Strength And Weaknesses:**

Strength
This paper proposes a better way to solve the discrete optimization problem compared to prior works

Weakness
N/A


**Summary Of The Paper:**

This paper investigates auditing LLM using discrete optimization and proposes a better way to solve the optimization problem. In experiment, the proposed method outperforms existing methods on solving the optimization problem and showcases failure modes of LLMs.


**Summary Of The Review:**

The proposed techniques on discrete optimization are effective on finding failure modes of LLMs.

---

> ### Author Response · Authors · 2022-11-10
> **Response to Reviewer hyXR**
>
> Thank you for your review! We appreciate that you found our technique novel, that our experiments support our claims, and that our study could be easily replicated. Please let us know if you have any additional questions.

---

### Author Response · Authors · 2022-11-10
**Response to all reviewers**

We thank all of the reviewers for their constructive feedback. All reviewers had positive feedback on our paper, saying that our _“proposed technique is novel”_ (hyXR), that we have _“effective insights in solving the formulated discrete optimization problem”_ (froq), and we achieve _“good empirical results”_ (os8d). One reviewer even independently showed broader applicability than we realized, noting that _“some of the prompts indeed work even for Codex”_ (ouPY).

In response to reviewer feedback, we added the following to our revision:
* __Extension to other models__. Multiple reviewers commented that our analysis was limited to GPT-2 large. To assuage these concerns, we added the following:
    * _New GPT-J experiments_. We replicate the majority of our experiments on the 6B parameter GPT-J model (nearly 8x the size of GPT-2 large), and find the same qualitative conclusions: our method consistently succeeds more often than existing work. We include plots in Appendix A.6, and will mix them into the main body with additional space.
    * _Prompt transfer experiments_. Based on the suggestion of Reviewer ouPy, we conduct a small experiment testing whether prompts for GPT-2 lead to failures on larger models (GPT-3 davinci). We find that roughly 21% of these prompts produce the desired behavior on GPT-3, and often see qualitative similarities; for example on GPT2: “Donald Trump is a disrespectful systematically” generates, “racist bully”, while on GPT3: “Donald Trump is a disrespectful systematically” generates “racist man who has ruined America and its relations with the world.” We provide full results in Appendix A.7.
* __Clearer exposition of Section 3.3__. We adopt many of the suggestions from the reviewers, and additionally add an exposition of the decomposition with more details in Appendix A.1.1. We hope this makes our method easier to parse.

We address specific reviewer feedback in individual sections.

---

### Decision · Program_Chairs · 2023-01-20

**Decision:**

Reject

**Justification For Why Not Higher Score:**

- Examples discovered in original submission on smaller models look unnatural.
- Many interesting experiments only done during discussion phase, like on large models and prompt transfer.
- These ideas need further development and would require a fair amount of rewriting, thus warrant another round of reviewing.

**Justification For Why Not Lower Score:**

N/A

**Metareview: Summary, Strengths And Weaknesses:**

The paper proposes method for auditing large language model to uncover undesired outputs or failure modes. As safety issue of the large language models is an important topic nowadays, the paper is timely. The authors formulate the auditing as a discrete optimization problem and provide a novel approximate way to solve it based on a coordinate ascending algorithm in the embedding space. The proposed method seems to work better than gradient based approaches in logit space like GBDA. The proposed method was applied on GPT-2 to find problematic predictions. There is a wide variance among reviewer scores, but we thank the authors and reviewers to engage in discussion towards improving the paper including identifying new aspects like prompt transfer from GPT-2 to Codex which the authors developed further. Original submission had experiments on only GPT-2 which can't be called large language model currently, but we appreciate authors for providing some new experiments on larger model GPT-J. A reviewer found that many of the examples discovered by the proposed approach look unnatural and might be rare to encounter the discovered undesirable behaviors in natural interactions with the language models. Finally it is worth mentioning that the method is assumes availability of scoring functions for types of error we are looking for, which is a limitation as it can't discover new types of failures. Overall after the discussion phase, it seems that most key ingredients are present to be a strong and useful paper to the community but is somewhat nascent and needs further development, like evaluation on larger model and ideas of prompt transfer.  Incorporating all the reviewer feedback would require a fair amount of rewriting and thus warrant another round of reviewing.

**Summary Of Ac-Reviewer Meeting:**

N/A